# Nanoparticle-Mediated Strategies for Enhanced Drug Penetration and Retention in the Airway Mucosa

**DOI:** 10.3390/pharmaceutics15102457

**Published:** 2023-10-13

**Authors:** Xin Yan, Xianyi Sha

**Affiliations:** 1Key Laboratory of Smart Drug Delivery, School of Pharmacy, Fudan University, Ministry of Education, Shanghai 201203, China; yanxin6767@163.com; 2The Institutes of Integrative Medicine of Fudan University, 120 Urumqi Middle Road, Shanghai 200040, China

**Keywords:** pharmaceutics, airway mucus, mucus penetration, mucoadhesion, nano-drug delivery systems, smart drug delivery system

## Abstract

Airway mucus is a complex viscoelastic gel composed mainly of water, glycoproteins, lipids, enzymes, minerals, etc. Among them, glycoproteins are the main factors determining mucus’s gel-like rheology. Airway mucus forms a protective barrier by secreting mucin, which represents a barrier for absorption, especially for more lipophilic drugs. It rapidly removes drugs from the airway through the physiological mucus clearance mechanism so drugs cannot remain in the lungs or reach the airway epithelial tissue for a long time. Significant progress has been made in enhancing drug lung deposition recently, but strategies are still needed to help drugs break through the lung mucosal barrier. Based on the physiopathological mechanisms of airway mucus, this paper reviews and summarizes strategies to enhance drug penetration and retention in the airway mucosa mediated by nano-delivery systems, including mucosal permeation systems, mucosal adhesion systems, and enzyme-modified delivery systems. On this basis, the potential and challenges of nano-delivery systems for improving airway mucus clearance are revealed. New ideas and approaches are provided for designing novel nano-delivery systems that effectively improve drug retention and penetration in the airway mucus layer.

## 1. Introduction

The mucus layer plays a vital role in human health, as it is the front line in the body’s defense system [1], capable of selective permeation of foreign substances and pathogens, thus protecting the proper functioning of the organism [2]. Pulmonary mucus comprises two layers: the liquid layer of the airway surface and the periciliary layer. The former consists of a gel-forming mucus responsible for the adsorption and encapsulation of inhaled particles. The latter is where the surface cells beat and relax, effectively transporting the mucus layer to the outside of the lungs [3,4]. The mucus layer’s composition and thickness are not constant; it is a dynamic system whose composition and thickness result from continuous secretion and clearance of mucus [5].

In addition to water, the main component of mucus is mucin, which can be divided into two subtypes based on its variability in glycosylation: secreted mucin and membrane-bound mucin. Membrane-bound proteins are mainly bound to the mucosal epithelial surface, while disulfide bonds between them link secreted mucins to form a continuous gel state [6]. Given their importance for lung drug bioavailability, studying lung mucus’s significant components and organization [5] is essential for understanding its barrier function. Direct drug delivery to the airway or inhalation therapy [7] is commonly used to treat lung diseases. Among them, inhalation therapy is more accessible to deliver the therapeutic drug to the lung mucosa and, therefore, can significantly reduce the drug’s dose and further reduce its side effects. Nevertheless, due to the pulmonary mucosa’s inherent clearance mechanism, the drug’s bioavailability in the lungs is low, and most of the drug is cleared out in the pulmonary mucosa [8,9].

Over the past decades, convincing data have confirmed that nano-delivery systems can be promising carriers for delivering drugs to penetrate the mucus layer, including polymeric nanoparticles, liposomes, polymeric micelles, and nanoparticles [10,11,12,13,14,15]. After the drug is delivered to the airway via inhalation, these nanoparticles can penetrate or remain in the airway mucus layer through permeation-mediated, adhesion-mediated, and bionic-mediated approaches, respectively [16,17,18]. The benefits of utilizing nanocarriers are enhanced drug bioavailability and reduced unwanted toxicity due to their surface modifications, suitable nano-size, and blood stability [19,20]. This paper outlines and summarizes new approaches of nano-delivery systems in improving drug retention and penetration in the airway mucus layer based on the potential and challenges of nano-delivery systems in improving airway mucus clearance. New ideas and approaches are provided to design novel nano-delivery systems that effectively improve drug retention and penetration in the airway mucus layer.

## 2. Pathophysiology of Airway Mucus

According to statistics, a typical adult breathes approximately 16–20 times per minute in a calm state. The amount of gas inhaled or exhaled each time is about 500 mL, which becomes the tidal volume [21]. Therefore, the human airway surface constantly interacts with the external environment, including inhaled particles and pathogens. The mucociliary clearance mechanism is an important defense mechanism of the human airway to maintain its normal physiological state [22]. The mucociliary clearance system avoids the retention of pathogenic bacteria by constantly renewing the mucus blanket while removing pathogenic microorganisms and inhaled particulate matter. The mucosal cilia clearance system [23] has three main components: the surface fluid layer in contact with the airway lumen, the periciliary fluid layer that supports cilia beating, and the respiratory epithelium, composed of secretory cells [24] (Figure 1).

The mucus layer is one of the crucial components of the mucociliary clearance system, which acts as both a physical and chemical barrier. The mucus layer comprises hundreds of substances, containing 98% water and 2% solids [25]. The major macromolecule of this 2% solid material is mucin, a large molecule formed by the highly glycosylated family of glycoproteins [26,27,28]. The major secreted mucins in the airways are MUC5B and MUC5AC [29,30], which have characteristic structural domains formed by repetitive tandem associations of abundant proline, serine, and serine and threonine. The repetitive sequences undergo O-glycosylation to stiffen the mucin backbone and increase the stiffness of the mucin chains, thus maintaining the gel morphology of the mucus. The mucin itself can bind to the therapeutic drug in a non-specific manner. Pavan G. Bhat and colleagues studied the permeability of porcine gastric mucus to five substances: isoniazid [31], Pentamidine [32], rifampicin [33], para-aminosalicylic acid [34] and pyrazinamide [35], all of which could be delivered to pulmonary targets using inhalation therapy. All five drugs showed significantly reduced permeability in the presence of mucus compared to the permeability of a blank buffer solution [2]. The above results suggest that all compounds bind specifically to mucin molecules before crossing the mucus layer, causing a reduction in their penetration through the mucus layer. The pore size of the mucus layer also plays an essential role in the permeation of drugs. Anionic and nonionic surfactants have a more significant effect on the nanoparticle mucus permeability and its mucus barrier modulation ability, and the mucus barrier modulation ability also depends on the surfactant type. Sodium dodecyl sulfate (SDS) increased the complex viscosity and viscoelasticity of the mucus, but poloxamer showed a decreasing trend. Tween 80 largely retained the original mucus rheological and morphological properties and may be a promising candidate for facilitating nanoparticle penetration of the mucus barrier with a favorable safety profile. Studies have shown that some small molecules can cross the mucus barrier by free diffusion, but most large molecules do not easily pass through the mucus barrier [27]. Therefore, we can infer that the permeability of the mucus layer may be limited by its pore size [36].

The periciliary liquid layer (PCL), ideal for cilia flow, is approximately 5–10 µm thick, [4] corresponding to the cilia’s length. If the layer is too thick, the cilia cannot reach the upper mucus layer and thus cannot perform their clearance defense function. At the same time, if the layer is too thin, the upper mucus layer will adhere to the cilia and prevent their movement [23]. Hydration of airway surface fluids is essential to achieve average mucus clearance [25]; in a normal airway [37], water is distributed between the mucus layer and the periciliary fluid layer, and the layer with low osmotic pressure is more likely to change its concentration than the layer with high osmotic pressure. Button et al. proposed a brush gel model that suggested that when the mucus layer was heavily hydrated [4], its osmotic pressure decreased dramatically so that fluid from the airway surface entered the mucus layer first while the PCL remained unchanged. Conversely, when airway dehydration occurred, the mucus layer was first dehydrated, increasing its concentration, which increased the osmotic pressure of the mucus layer, and the PCL layer was compressed under high pressure, leading to PCL collapse. PCL is a mesh structure composed of various macromolecular substances, the size of which is not constant between the meshes and is related to the height of the PCL layers [38]. When excessive hydration of the airway surface fluid occurs, resulting in the collapse of the PCL layer, the grid size of the PCL layer is subsequently reduced; assuming that the drug particles reach the mucus layer, the penetration of the drug particles in the pulmonary mucosa is decreased significantly due to the reduction in the PCL grid voids. In addition, the composition and structure of the mucus layer undergo a series of changes in pathological states, such as an imbalance in ion transport in the pulmonary airways of patients with cystic Fibrosis (CF), leading to a reduction in airway surface fluid volume, a marked increase in mucus viscoelasticity, and impaired clearance of mucus cilia [39,40,41,42]. In addition, under pathological conditions, the concentration of mucin, DNA, and actin is significantly increased, significantly reducing the average size and size distribution of the web spacing, thus severely hindering the transport of nanoparticles. Thus, there is an urgent need for nano-delivery carriers that can carry drugs through the dense mucus layer.

## 3. Nanoparticle-Mediated Effective Enhancement of Drug Retention and Penetration in the Airway Mucosa

Based on the previously described barrier properties of airway mucus, we can design different strategies to enhance drug penetration in the pulmonary mucosa based on its properties. Among them, nanoparticle formulations offer significant advantages in improving cell penetration and thus could be a promising approach for treating lung diseases [43]. However, when used as a transport carrier, it faces the double barrier of size filtration and interaction filtration of the mucus layer. Since the barrier properties of mucus change its behavior, a suitable design of nanoparticles is required, such as modifying the surface properties of nanoparticles (including particle surface functional groups and charge density, etc.), changing their particle size [44], etc. Furthermore, enlarging the mucus lattice void by disrupting specific non-covalent interactions of the mucus gel is also an effective method to promote mucosal drug penetration. Table 1 summarizes the various types of nanoparticles that enhance mucus penetration and retention.

### 3.1. Mucosal Permeation System

#### 3.1.1. Nanoparticles Based on Hydrophilic Polymer Modifications

For diseases of the lungs, the use of inhalation therapy to deliver drugs to the lungs is a standard treatment method. Due to the prevalence of sialic acid and sulfate groups, mucus gels have a net anionic charge [60]. Still, carriers loaded with therapeutic drugs are usually positively charged, making it difficult for medicinal medications to penetrate the mucus layer of the lung. They are often adsorbed by the mucus layer and removed by the mucosal cilia clearance system before reaching the drug target of the target cells [61]. Therefore, neutral, uncharged nanoparticles are more likely to penetrate the mucus. Given these limitations, various hydrophilic polymer-modified nanoparticles, including polyethylene glycolic nanoparticles, nanoparticles composed of poly(acrylic acid) (PAA) and poly(allylamine) (PAM), and carboxy-modified polystyrene nanoparticles, have been carefully designed to overcome the airway mucus barrier and prolong the retention time of drugs in the mucus layer [47]. Among them, the nanoparticles with surface-coated polyethylene glycol have a net neutral charge on the surface, which does not interact with mucin by charge adsorption. The penetration rate in airway mucus is substantially increased [62,63,64] (Figure 2).

One of the resistance mechanisms to Pseudomonas aeruginosa infection is the formation and presence of biofilms, and P. aeruginosa is a significant pathogen of diseases such as persistent pulmonary infections in patients with cystic Fibrosis (CF) [65,66,67]. Bahamondez-Canas and colleagues coupled tobramycin with PEG to determine its in vitro activity against Pseudomonas aeruginosa. They found that polyethylene glycosylated tobramycin was significantly superior to free tobramycin in reducing Pseudomonas aeruginosa proliferation in cystic fibrosis mucus barriers [68]. In another work, Kolte and colleagues report on polyethylene glycolic PLGA and polyethyleneimine composite nanoparticles for pulmonary delivery of pDNA. The polyethylene glycolic complex particles were converted to DPI by a lyophilization technique and bound to lactose carrier particles, resulting in improved nebulization performance and lung deposition [45]. They also studied the number of composite particles diffusing the mucus layer before reaching the lung epithelium through mucus permeation experiments. They found that in the case of PEG-modified composite particles, a more significant proportion of the composite particles diffused more in the gelatin than in the case of non-polyethylene glycolic composite particles. Lung cancer is one of the most common cancers worldwide [69,70], and in this context, chemotherapeutic agents hold great promise for cancer treatment. Still, the challenge is delivering chemotherapeutic agents to low-dose drug targets. Inspired by nano-delivery carriers, Krishna Rao and colleagues prepared nanoparticles from methoxy polyethylene glycol (mPEG) and adriamycin (DOX) complexes. In an ex vivo lung cancer model, PEG’s molecular weight had a substantial effect on the rate and extent of cellular internalization of DOX and in vitro cytotoxicity. These nanoparticles were easily dispersed in a propellant-based dosing inhaler, and an aerosol size favorable for deep lung deposition (FPF 66%) was obtained [46]. However, studies have shown that some anti-polyethylene glycol antibodies exist in humans. After delivering polyethylene glycolic particles to the human body, these particles tend to develop immune reactions with the human body [71]. Therefore, the feasibility and safety of polyethylene glycolic drug carriers remain a crucial consideration. In addition, no single surface chemistry allows drug carriers to break both the mucus and cellular barriers, and polyethylene glycolic drug carriers are no exception to this rule. It was found that polyethylene glycolic drug carriers tend to impede cellular uptake, so there is a need to find chemicals that can replace polyethylene glycols in penetrating the mucus barrier. At the same time, their safety in humans is reliable.

In addition to polyethylene glycol, other hydrophilic polymers such as polyacrylic acid, polyallylamine, and carboxy-modified polystyrene have been reported in preliminary studies for modifying nanoparticles to enhance their mucus penetration ability. To improve the efficacy of the nano-delivery system, Laffleur and colleagues developed nanoparticles composed of polyacrylic acid (PAA) and polyallylamine (PAM), which diffuse 2.5 and 18 times more efficiently in mucus than PAM and P.A.A. NPs alone, respectively [47]. In another study, Dawson and colleagues found that some particles whose surfaces were modified with carboxyl or amine groups were mobile in CF sputum. However, the pooled average transport rate was still at least 300 times slower than the identical particles in water [40]. The nanoparticles modified by hydrophilic polymers effectively increased penetration ability in the mucus layer and enhanced stability in the mucus layer.

#### 3.1.2. Nanoparticles Based on Cell-Penetrating Peptide Modifications

A high mucus layer intercepts foreign pathogens and simultaneously prevents most model drugs and polymeric nanoparticles from penetrating, which is required for successful treatment. Thus, drug penetration of the mucus barrier and reaching epithelial cells is a long-term challenge. Cell-penetrating peptides can facilitate drug and particle mucosal delivery and potentially promote cellular uptake [72] (Figure 3). Leal and coworkers measured the transport behavior of mucus-penetrating peptides complexed with nanoparticles in a co-culture assay of lung epithelial cells from CF patients. They found that the selection of mucus-penetrating peptides improved the bulk diffusion of phage in the CF sputum of patients by approximately 600-fold compared to a positively charged control phage clone and enhanced cellular uptake of conjugated nanoparticles by a factor of 3 compared to unmodified carboxylated and mPEG-conjugated nanoparticles [48]. Thus, cell-penetrating peptides can be effectively used as surface modifications of drug carriers, potentially increasing the number of drugs and drug carriers delivered to lung epithelial cells and thus improving clinical outcomes. Another research work demonstrated a novel nanocomposite consisting of supramolecular cell-penetrating peptides that formed fibers and were coated on poly(lactic acid-glycolic acid) (PLGA) nanoparticles to enhance lung drug delivery. The results showed that these nanocomposites showed a threefold increase in intracellular delivery of nanoparticles in various cells (including primary lung epithelial cells and macrophages) compared to bare PLGA nanoparticles [73]. The nanocomposites also showed effective mucus penetration, and the nanocomposite powders were freeze-dried and nebulized without affecting their physicochemical and biological activities. To make cell-penetrating peptides more efficient for drug delivery in vivo, further polyethylene localization of cell-penetrating peptide nanoparticles has been reported to enable drug carriers with specific mucosal penetration ability and cell permeation potential. Based on the above concept, Osman and colleagues prepared a novel cell-penetrating peptide (CPP)-a based non-viral vector that directly couples the peptide to DNA via electrostatic interactions to form nanoparticles and to adapt the GET peptide for efficient in vivo delivery. They designed a polyethylene glycolic version of the peptide. When tested in an in vivo mouse lung model, these particles showed excellent lung distribution and transgene expression compared to the non-polyethylene glycolic version while maintaining a good safety profile [1] (Figure 2). GET peptides have great potential for in vitro and in vivo gene transfer. The peptide-coated nanoparticles address multiple delivery barriers and may effectively serve as an excellent alternative to standard PEG surface chemicals, promising improved therapeutic outcomes in mucus-obstructive lung disease.

#### 3.1.3. Lipid-Based Nanoparticles

Lipid nanocarriers include nanoemulsions, liposomes, solid lipid nanoparticles (SLNs), nanostructured lipid carriers (NLCs), lipid nanocapsules (LNCs), lipid–polymer hybrid nanoparticles (hybrids), and high-density lipoproteins (HDLs). Lipid nanocarriers have excellent physicochemical properties such as cellular affinity and biocompatibility, and their application in airway mucosal drug delivery has been widely explored. Good efficacy has been reported using pulmonary lipid carriers for treating asthma and tuberculosis [10]. Liposomes are amphiphilic and are excellent carriers of drugs. In airway mucosal drug delivery, liposomes also significantly improve the therapeutic effect. Gupta and coworkers encapsulated fasudil (a rho-kinase inhibitor) into liposomal vesicles for the treatment of pulmonary hypertension [49] (Figure 4), and when liposomal fasudil was given as an aerosol, mean pulmonary artery pressure (MPAP) decreased by 37.6 ± 5.7% compared with intravenous fasudil and continued to decrease for approximately three hours.

In addition to liposomes, phospholipid complexes have strong cellular affinity due to their cell membrane-like structure. They can improve the stability of phospholipid complexes when co-applied with self-nano emulsification systems (SNEDDS) [74,75]. Self-nano emulsifying drug delivery systems (SNEDDS) have the added advantage of avoiding adhesion to mucus mucins, improving the bioavailability of hydrophobic drugs, and allowing them to reach epithelial cells through the mucus layer. This system is typically used to deliver hydrophobic drugs, but hydrophilic drug complexes with amphiphilic fractions can also be incorporated into a self-nano-emulsifying drug delivery system [76]. The self-emulsifying drug delivery system is an automatic O/W system composed of water [75], oil, and surfactant, and the resulting droplet size is approximately 100–200 nm. Its surface is hydrophobic and therefore interacts less with mucus mucin. Two of the most critical factors affecting the mucus permeation behavior of this system are the particle size of the various formulations and the composition of the formulations. Several SNEDD formulations were designed and evaluated for uniformity, stability, and particle size. The selected formulations resulted in particle sizes ranging from 5 to 455 nm. Mucus diffusion studies showed self-nano emulsifying drug delivery systems to show size-dependent penetration [77], with the smaller the droplet size, the greater the mucus penetration. The diffusion of the 4% permeates formulation increased 70.19 times in 8 h for a formulation particle size of 12 nm. The permeate diffusion was much lower for a formulation particle size of 40 nm maximum. In addition, the type and concentration of an excipient significantly affected the permeation behavior of SNEDDS. Increasing the concentration of Cremophor RH 40 from 20% to 30% or up to 40% in SNEDD formulations resulted in a twofold and fourfold increase in droplet permeation behavior, respectively. Further increases in Cremophor RH 40 to 50–80% did not increase mucus penetration. In another study, Karamanidou and colleagues developed a novel mucus-permeable SNEDDS system formulated with hydrophobic ion-pair insulin/dimyristoyl phosphatidylglycerol, characterized by an average droplet diameter of 30–45 nm, for insulin delivery. In vitro, permeability results showed that SNEDDSc exhibited more mucus permeation than SNEDDSa and b. The above results may be attributed to a potential interaction between mucus and SNEDDS and that increased amounts of long-chain triglycerides lead to enhanced mucus permeability [50].

Small interfering RNAs hold great promise in treating lung diseases, such as cystic fibrosis [78,79]. However, despite the tremendous therapeutic potential of small interfering RNAs, translating inhaled small interfering PNAs from experimental to clinical settings remains difficult. The key to fully exploiting the potential of small interfering RNAs lies in preparing safe and effective nano-delivery systems that carry small interfering RNAs to reach pulmonary therapeutic targets [80]. Lipid polymer hybridized nanoparticles are a promising drug delivery system that can help nucleic acid cargo get pulmonary marks through the mucus-covered human airway epithelial barrier. Conte and coworkers prepared lipopolymer hybrid nanoparticles consisting of a poly(lactic acid-glycolic acid) (PLGA) core and a dipalmitoylphosphatidylcholine (DPPC) lipid shell for delivery of nucleic acid drugs to the lung. They investigated whether the mucus penetration ability of this nanocarrier was altered after polyethylene globalization [51]. Overall, lipid nanoparticles, as a non-invasive route of airway mucosal drug delivery, can improve the therapeutic efficacy of most drugs. In addition, lipid nanoparticles are expected to be innovated with state-of-the-art inhalation instruments for the clinical treatment of respiratory diseases. In contrast, the development of lipid nanoparticles at scale is expected to progress.

### 3.2. Mucosal Adhesion System

The process by which mucosal adhesion occurs between the mucosal adhesion polymer and the mucus layer is relatively complex, and the theoretical mechanisms that explain this process include adsorption, diffusion, wetting, and electrostatic forces. In adsorption theory, mucus adsorption is caused mainly by ionic, covalent, and metal bonds. Secondary interactions arising from hydrogen bonding, van der Waals, electrostatic forces, etc., also produce mucus adsorption. Diffusion theory describes the interpenetration of polymers in different regions of mucin, which is influenced by the molecular weight of the chains, their cross-linking, and their mobility and flexibility. Longer polymer chains allow greater diffusion and penetration of mucus. When the polymer and mucus have good mutual solubility and structural similarity, the polymer can diffuse better in the mucus [81]. The wetting theory applies primarily to liquid or low-viscosity bioadhesives and represents the spreading of the polymer on the adhesive surface. For successful retention in the mucus, these polymers should overcome interfacial tension and diffuse spontaneously through the mucus. In general, the structural and functional group similarities between the adhesive polymer and the mucilage increase the vehicle miscibility of the two, resulting in a high spreading of the polymer in the mucilage layer. In such systems, the low contact angle between water and the polymer enhances the hydration of the polymer chains, thus facilitating the contact between the polymer and the mucus layer.

The process of mucus adhesion can be roughly divided into two steps: Firstly, close contact between the drug carrier and the mucus layer occurs, and the drug carrier absorbs a certain amount of water from the mucus layer, increasing the retention time in the mucus layer; then the drug carrier enters into the mesh structure of the mucus layer and interacts with the mucus layer in a covalent and charge-directed electrostatic adhesion [82].

#### 3.2.1. Nanoparticles Based on Chitosan Modification

Chitosan is formed by the deacetylation of chitin in an alkaline environment and is one of the most versatile natural polymers because of its biodegradability [83], biocompatibility, and low toxicity. Because its primary amine is protonated and positively charged, it can efficiently react with anionically seized substances to form complexes. In the pulmonary mucosa, due to the negative charge of mucus gel, when the chitosan-based drug carrier arrives, the natural strong electrostatic adhesion of chitosan to the mucus layer occurs [84,85,86,87], which endows chitosan with particular mucosal adhesion ability and increases the retention time of the drug in the pulmonary mucosa, resulting in slow and gradual release and absorption of the drug, facilitating the drug to reach the pulmonary epithelium at the right time [87]. Chitosan produces free amino groups during deacetylation, and the negatively charged mucus layer interacts with the positively charged amino groups; the higher the number of free amino groups, the stronger the adhesion ability of chitosan in the pulmonary mucosa. Thus, the adhesion capacity of chitosan in lung mucosa increases with the degree of deacetylation. However, unmodified chitosan has some inherent disadvantages, including a high degree of hydrolysis in an acidic environment, no inhibition of some hydrolytic enzymes in the mucus layer, and a tendency to be removed quickly in the mucus layer [88]. An effective strategy is to derivatize chitosan by modifying chitosan [89], such as thionation and acylation so that the retention time of chitosan-based drug carriers in the mucus layer is prolonged.

##### Nanoparticles Modified with Thiolated Chitosan

Thio-chitosan is a compound formed by immobilizing thiol groups on the primary amine group of chitosan [90]. When it reaches the mucus layer, it reacts with the mucus to form disulfide bonds between the two, thus adhering to the mucous membrane. The delivery potential in the gastric retention system was determined by preparing chitosan-4-isobutyl tablets. After evaluating the adhesion properties of the tablets in the gastric mucosa using the TA-XTplus mass spectrometer, the results showed that chitosan-4-isobutyl tablets require a 1.533-fold higher vertical separation force than unmodified tablets to break the mucosal adhesion bonds. This study suggests that thiol chitosan may be a promising formulation for mucosal delivery systems.

Dünnhaupt and colleagues investigated the distribution behavior of Thio-chitosan nanoparticles on the intestinal mucosa. They observed the mucus penetration properties of unmodified and modified chitosan nanoparticles in vitro and in vivo. The results showed that the modified particles showed more than 6-fold adhesion than the original [54]. This has important implications for optimizing chitosan nanoparticles and must be designed rationally to achieve the desired effect. In another study, Lichen Yin and colleagues further demonstrated the adhesion ability of the nanoparticles [91] (Figure 5). They synthesized trimethyl chitosan-cysteine coupling and formed polyelectrolyte nanoparticles with insulin by self-assembly, which showed a significant increase in mucus adhesion compared to insulin nanoparticles, which may be partly attributed to the formation of disulfide bonds with mucin; in addition, the modified nanoparticles resulted in increased insulin cell internalization. Therefore, self-assembled nanoparticles between trimethyl chitosan-cysteine coupling and protein drugs may be an effective and safe mucosal delivery system [91]. Based on the above studies, thiol chitosan NPs profoundly affect mucosal adhesion properties, crucial for inducing in vitro or in vivo effects.

##### Nanoparticles Modified with Quaternized Chitosan

The quaternization of chitosan is carried out on its amino and hydroxyl groups, mainly by introducing quaternary ammonium salts on the amino groups of chitosan or the amino groups of chitosan derivatives such as N, N, N-trimethyl chitosan [92]. Quaternary ammonium salts are usually characterized by high site resistance and strong hydration capacity, and the introduction of quaternary ammonium salts into chitosan molecules can weaken the intra- and intermolecular hydrogen bonding of chitosan molecules and improve the solubility of chitosan derivatives. It has a wide range of application prospects [93,94]. N, N, N-trimethyl chitosan chloride nanoparticles, after mannosylation, can be used as a novel pulmonary drug delivery vehicle for the pulmonary delivery of drugs. Etoxifylline, a bronchodilator, is widely used in various airway diseases, and its incomplete absorption and significant first-pass effect lead to poor bioavailability of its tablet and injectable forms. Based on the above aspects, preparing mannosylated TMC using ionic gel technology with sodium tripolyphosphate (TPP) as an ionic cross-linking agent can effectively avoid hepatic first-pass metabolism, enhance pharmacokinetic analysis, and reduce off-target side effects through pulmonary administration [55] (Figure 6). Studies have shown that the degree of quaternization of chitosan has a relatively significant impact on mucosal adhesion and mucosal permeation [95] and that the degree of quaternization of chitosan determines its surface charge density, especially in neutral or alkaline environments. Thanou and colleagues synthesized trimethyl chitosan chloride with 40% and 60% substitution and studied their effect on the permeability of intestinal Caco-2 monomolecular membranes with tight junctions. They found that the membrane permeability of trimethyl chitosan chloride with 60% substitution was higher than that of 40% substitution at all concentrations. These results suggest that high charge density is necessary for chitosan derivatives to improve mucosal membrane permeability significantly [96]. However, their results only demonstrated that the increased quaternization of chitosan derivatives is beneficial for mucosal adhesion, but whether the higher quaternization of chitosan derivatives is better was not confirmed. This issue was further explored by Hamman et al. To determine the optimal degree of quaternization, they synthesized TMC polymers with different degrees of quaternization. They evaluated their absorption enhancement properties through rat nasal epithelium at pH values of 6.20 and 7.40. The results showed that the uptake performance of the polymers increased with increasing quaternization, and the optimal quaternization reached a maximum at pH 7.40, which did not increase the uptake of the compounds even with subsequent increases in quaternization [97].

#### 3.2.2. Hyaluronic Acid-Based Modified Nanoparticles

Hyaluronic acid (HA) is a non-sulfated glycosaminoglycan widely found in nature and has biocompatible, biodegradable, and viscoelastic properties that offer potential advantages in developing bioadhesive drug delivery systems [98,99]. Some researchers have found that people with COPD can increase bronchial patency with multiple inhalations of hyaluronic acid [57]. Salbutamol sulfate is an adrenergic agonist widely used in bronchial diseases [100]. After pulmonary administration via a dry powder inhaler, salbutamol has a short retention time in the pulmonary mucus layer due to its high water solubility. Therefore, encapsulation of salbutamol sulfate in dry powder form in inhalable HA pellets can be considered to improve the retention time of the drug in the pulmonary mucus layer by taking advantage of the bioadhesive properties of HA. When salbutamol was made into particles with HA, the retention of local salbutamol in the lungs was more than three times higher than salbutamol without HA throughout the period. In addition, the residence time of salbutamol was significantly prolonged from 2 h to 8 h with the help of hyaluronic acid [57].

However, for poorly water-soluble drugs, hyaluronic acid is still available to prolong the retention time of the drug in the mucus layer through its bioadhesive effect. Budesonide is a poorly water-soluble corticosteroid with vigorous anti-inflammatory activity. It has been shown that budesonide was incorporated into inhalable hyaluronic acid pellets after ray drying, prolonging the pharmacological action time of the drug. In vitro release tests showed that the pharmacological effects of inhaled particles were significantly extended in rats compared to inhaled budesonide nanocrystal suspensions [58]. This can be attributed to the mucosal adhesion of the polymer, which overcomes mucociliary clearance and prolongs the active substance’s retention in the lungs without reducing the in vitro dissolution rate. In the lung, high molecular weight hyaluronic acid exerts anti-inflammatory and anti-angiogenic effects, promotes cell survival, and enhances adhesion [101]. Also, the conformation of hyaluronic acid molecules strongly influences the mucoadhesive properties of hyaluronic acid nanoparticles. Rouse and coworkers further demonstrated the effect of hyaluronic acid on hydrophobic drugs in mucus adhesion by investigating the interaction and adsorption of the hydrophilic polymer hyaluronic acid (HA) on the surface of the hydrophobic corticosteroid drug fluticasone propionate (FP). It was shown that the mucoadhesive ability of nanoparticles was significantly higher when the hyaluronic acid molecules were in the secondary structure compared to the tertiary network structure [102]. Based on the above studies, hyaluronic acid has an essential role in the development of bioadhesive drug delivery systems. By rational design of the molecular weight of hyaluronic acid and its molecular conformation, the retention of hyaluronic acid nanoparticles in airway mucus can be promoted, and the ability to recognize specific receptors overexpressed on the surface of tumor cells can be improved.

### 3.3. Mucolytic Agent Modified Nanoparticles

Airway mucosal drug delivery not only allows for local targeting but also offers the possibility of systemic drug targeting. The mucus layer of the airway mucosa is a challenging barrier and is a fundamental cause of low drug bioavailability. Based on the above strategies to enhance drug retention and penetration in the airway mucosa, in addition to the mucosal permeation system and mucosal adhesion system described above, drug penetration can also be facilitated by mucolytic agents modifying nanoparticles to locally cut the mucin network structure to alter the intrinsic properties of mucus. Mucolytic enzymes are essential substances that break down the three-dimensional structure of mucus and can reduce the viscous and elastic properties of the mucus layer by breaking down the internal structure of mucus. Mucolytic enzymes can be broadly classified into two types: thiols with free sulfhydryl groups that cleave the disulfide bonds between mucus mucins and peptide bonds that break the amino acid sequence of mucins [101].

#### 3.3.1. Nanoparticles Based on Thiol Modification Containing Free Sulfhydryl Groups

Studies have shown that N-acetylcysteine (NAC) has significant efficacy as a mucolytic agent in overcoming the airway mucus barrier and is used for administration to the airways to relax secretions. It reduces the degree of cross-linking between mucins mainly by cleaving the disulfide bonds between mucins [103]. Therefore, NAC encapsulated in nanoparticles can be considered for the treatment of lung diseases. NAC is usually administered at 400–1200 mg daily as a mucolytic agent. After oral administration, the drug is absorbed quickly but tends to undergo first-pass metabolism, and NAC can also be administered by inhalation to treat pulmonary disease. Martin and colleagues investigated the effects of several sulfhydryl groups on the rheology and mucociliary transport properties of recombinant canine tracheal mucus model secretions. They found that N-acetylcysteine reduced the mucus elastic modulus and thus improved mucociliary transport at concentrations where mucin did not precipitate. In contrast, s-carboxymethyl cysteine did not affect either mucus properties or mucociliary transport rates. Pulmonary fibrosis is a chronic inflammatory interstitial lung disease with pathological symptoms of mucus gland hypertrophy and excessive airway mucus secretion. Studies suggest that NAC may be beneficial in this disease. The area of pulmonary fibrosis, the number of mucus-secreting cells in the airway epithelium, and the expression of Muc5ac messenger ribonucleic acid and protein were significantly reduced by morphological analysis after NAC administration to rat lungs [104].

#### 3.3.2. Protein Hydrolase-Based Modified Nanoparticles

Protein hydrolases enable local disruption of the mucus layer by cleaving peptide bonds in the amino acid sequence of mucin. The hydrolysis efficiency of various proteases varies at different pH values. For example, while trypsin and pancreatic rennet are less suitable for use in the small intestine (pH 5.5–7.5), pepsin with an effective pH below 3.5 may be ideal for use in the vagina or stomach [105]. However, recombinant human deoxyribonuclease is more suitable for the respiratory tract. A large number of clinical data prove that nanoparticles carrying recombinant human deoxyribonuclease can be widely used in various lung diseases by fragmenting DNA molecules after nebulized drug delivery. It has been shown that inhalation of recombinant human deoxyribonuclease in patients with cystic fibrosis enhanced the clearance of tenacious airway surface secretions and improved lung function in patients with cystic fibrosis [106].

In conclusion, enzyme-modified nanoparticles have great potential as carriers for airway mucosal delivery, especially their ability to cleave the local mucus network and play an essential role in delivering drugs with low bioavailability. However, enzyme-modified nanocarriers also have some potential risks; the enzymatic breakdown of the mucus network structure increases the risk of some invaders, such as pathogens coming into contact with the airway epithelium, while the mucus renews in real-time, so the mucus layer has a limited time to be disrupted. In the future, we can modify nanoparticles by developing complex enzymes to improve their permeability and safety.

### 3.4. Clinical Study of Drug Delivery Targeting Mucus

Inhalation administration is one of the effective routes of administration for targeting respiratory diseases. The design of drug delivery systems must adequately consider the barrier function of airway mucus (Table 2). In addition, the permeability of particles through airway mucus depends on the microstructural organization of the mucus, its size, and the surface chemistry of the particulate matter. Sinko and coworkers studied the diffusion of variously sized polystyrene particles in a reconstituted porcine gastric mucin gel sandwiched between the two chambers of a Transwell-Snapwell diffusion chamber. They observed a sharp decrease in translocation permeability when particle sizes approached 300 nm. Sanders et al. reported that a deficient percentage of carboxylated polystyrene nanospheres moved through a 220 μm-thick CF sputum layer after 150 min, with the largest nanospheres studied (560 nm) almost entirely blocked by the sputum [107]. Cystic fibrosis has been a target for gene therapy since the cloning of the CFTR gene in 1989. Lung disease is the leading cause of morbidity and mortality in cystic fibrosis patients, with a median age of death of 29 years. A double-anonymized, placebo-controlled trial (*n* = 8 cystic fibrosis patients given DNA-lipid complexes alone, *n* = 8 control patients given lipids/placebo alone) found that pulmonary administration of CFTR DNA-lipid complexes (but not placebo/lipids alone) significantly corrected chloride abnormalities identified by measurement of in vivo potential difference and chloride efflux [108]. Peadar G. Noone and colleagues Noone and colleagues studied 11 adult CF patients in a study protocol that allowed comparisons in a single subject: vector and placebo were sprayed into alternate nostrils at intervals over seven hours. After dosing, vector-specific DNA was present in nasal lavage of all subjects for up to 10 days. There were no adverse events. The vector-treated epithelium did not exhibit a significant increase in CFTR-mediated Cl-conductance from baseline and was not different from the placebo-treated nostril: mean ΔCFTR Cl-conductance, mV ± SEM, −1.6 ± 0.4 vs. −0.6 ± 0.4, respectively. CFTR-mediated Cl-conductance increased toward normal during repetitive nasal potential difference measurements over the three days before dosing, influencing postdosing calculations. These studies demonstrate the potential of drug delivery systems targeting mucus and can be formulated with various drug delivery systems to target respiratory diseases such as pulmonary and cystic fibrosis. More clinical trials are needed to validate the advantages of these advanced drug delivery systems (e.g., nanoparticles) over traditional drug delivery methods [109].

## 4. Conclusions and Perspectives

When the conventional drug delivery system is faced with the clearance mechanism of the airway mucosa, the airway mucus can hinder the delivery of drugs from the lungs, resulting in a large amount of drug loss and not achieving the best therapeutic effect. This condition can lead to a significant increase in the drug’s minimum inhibitory concentration and increase the administration frequency, which can affect patient compliance. Therefore, developing novel drug delivery systems is necessary to extend the retention time of drugs at the mucosal site to absorb medicines better. In addition, choosing a biodegradable and biocompatible delivery system for lung delivery across the mucus barrier ensures that drug release is controlled according to the system’s biodegradation rate. In this review, we discuss several novel nanoparticle delivery systems where particle size, surface morphology, surface charge, and targeted molecules can all be altered, as these properties are critical for differentiating the slime adhesion or slime permeability of nanoparticles. Compared with uncoated nanoparticles, nanoparticles coated with PEG, chitosan, hyaluronic acid, and surfactants have higher permeability and better pharmacokinetic characteristics and can extend the residence time of the drug in the lungs. In addition, when the particle size of the nanoparticles is smaller than the mesh size of the mucus, it is easy to diffuse through the mucus, and the absorption rate of the drug is higher in the lungs, which is an effective alternative to traditional therapies.

Nanoparticles may then have some potential toxicity when inhaled. When nanoparticles enter the body through inhalation, the lungs are one of the main ways for NPs to enter the body, and, therefore, the possible sites for NPs accumulation. Once nanoparticles enter the interstitial space of the lung and are rapidly absorbed by alveolar cells, they can induce toxic effects, such as the generation of oxidative stress, DNA damage, and inflammation, leading to fibrosis and pneumoconiosis. Inhalable particles cause an increase in reactive oxygen species, which can lead to oxidative stress. The following factors may cause the production of ROS: (1) active REDOX cycle reaction occurs on the surface of nanoparticles, especially on the surface of metal nanoparticles; (2) oxi-dative groups functionalized on NPs; and (iii) particle–cell interactions, especially in the lungs where there is a rich pool of ROS producers like the inflammatory phagocytes, neutrophils, and macrophages. In addition, nanoparticles may lead to the development of genotoxicity, and it is believed that there is a more significant potential for damage when NPs can get close to DNA. Other different mechanisms may be specific to the elemental composition and shape of NPs, which could lead to DNA damage such as single-strand breaks, double-strand breaks, DNA deletions, and genomic instability in the form of increase in 8-hydroxy-20-deoxyguanosine levels. In addition, under pathological conditions, the concentration of mucin, DNA, and actin is significantly increased, which substantially reduces the average size and size distribution of the web spacing, thus severely hindering the transport of nanoparticles, hence the urgent need for nano delivery carriers that can carry drugs through the dense mucus layer. Inspired by nanoparticles’ superior mucus penetration and adhesion properties, researchers have designed various functional nanoparticle-mediated novel strategies for enhancing drug penetration and retention in the airway mucosa. These nanoparticles mainly include mucus-penetrating, mucus-adhesive, and mucolytic agent-based nanoparticles. These nanoparticles can overcome the limitations of modern pulmonary drug delivery, improve drug residence time and mucosal penetration in the mucus layer, promote drug absorption, and better carry the drug to the target site, and have made some progress in preclinical trials for effective enhancement of drug retention and penetration in the mucus layer.

Advances in the nanoparticle-mediated effective therapeutic enhancement of drug penetration and retention in airway mucus are mainly reflected in the following aspects: (i) enhanced mucus penetration and mucus adhesion of therapeutic drugs; (ii) reduced toxic effects of therapeutic drugs; (iii) improved cellular affinity and biocompatibility of therapeutic drugs; and (ii) ability to co-deliver drugs with different targets into the same nanoformulation for synergistic effects. Thus, nano-delivery systems are promising for clinical development in enhancing drug mucus penetration and retention. However, challenges still make clinical translation more difficult to achieve. Under pathological conditions, the composition and structure of the airway mucus layer are variable, highlighted by a decrease in airway surface fluid volume, a significant increase in mucus viscoelasticity, and impaired mucus cilia clearance. Although the mucus penetration of nanoparticles can be enhanced by surface chemical modification or biomimetic means, many nanoparticles are still captured by the mucus clearance system. They cannot remain in the mucus layer for a long time. Further research on the airway mucus barrier is needed.

In summary, there is still a large gap between preclinical trials and clinical applications of nano-delivery systems in enhancing drug retention and penetration in the mucus layer. Considering the changes in the composition and structure of the airway mucus layer under pathological conditions, nanoparticles need to be rationally designed according to the changes in the mucus layer, thus enhancing their targeted delivery to the lesion site. With the development of nano-delivery systems and in-depth studies of the airway mucus layer in pathological states, novel nano-delivery systems need further development to facilitate their clinical translation in enhancing drug mucus penetration and retention.

## Figures and Tables

**Figure 1 pharmaceutics-15-02457-f001:**
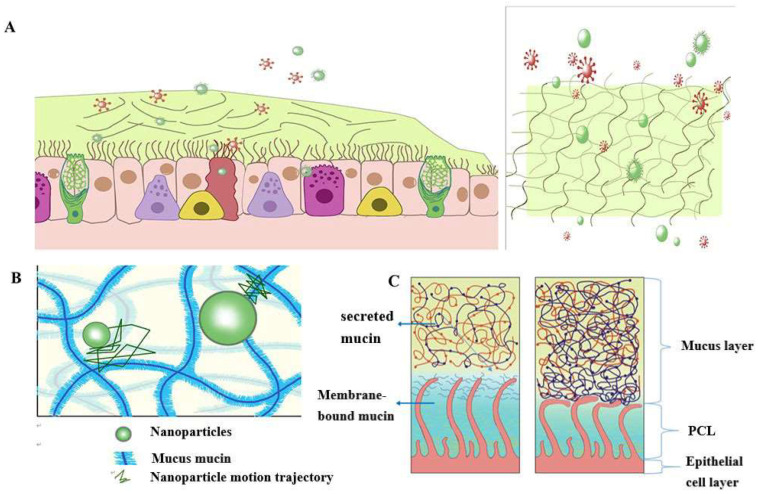
Schematic diagram of the structure and function of the mucus barrier: (**A**) The mucosal cilia clearance system has three main components: the surface fluid layer in contact with the airway lumen, the periciliary fluid layer that supports cilia beating, and the respiratory epithelium, composed of secretory cells. Constantly renewing the mucus blanket avoids the retention of pathogenic bacteria while removing pathogenic microorganisms (red spheres) and inhaled particulate matter (green spheres); (**B**) Schematic diagram of nanoparticle structure through mucin network. Small molecules can cross the mucus barrier by free diffusion, but most large molecules do not readily cross the mucus barrier; (**C**) Schematic diagram of PCL collapse due to dehydration of the mucus and PCL layers and the airway under normal conditions.

**Figure 2 pharmaceutics-15-02457-f002:**
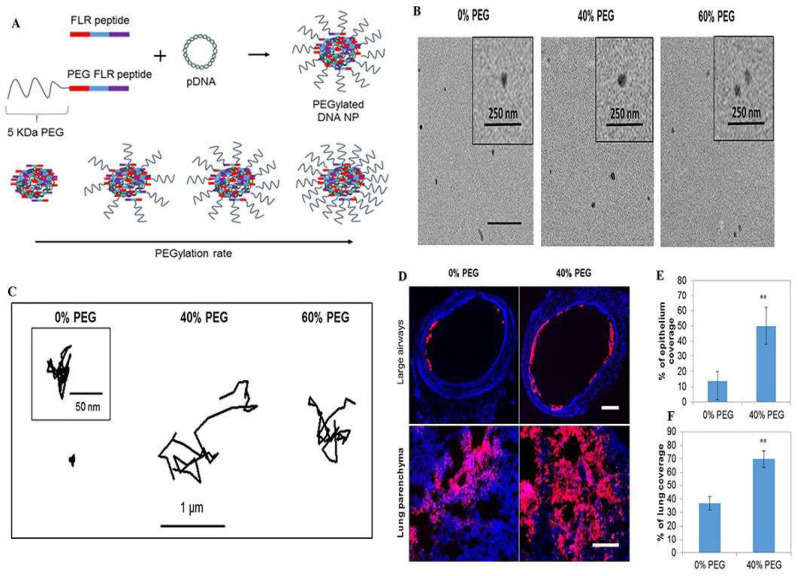
PEGylated enhanced cell-penetrating peptide nanoparticles for lung gene therapy: (**A**) Schematic of non-PEGylated and PEGylated FLR peptides blended at different molar ratios to form D.N.A. NPs with a tuneable coating of PEG on the outer surface; (**B**) Representative transmission electron micrographs of the respective NPs; (**C**) Representative trajectories of particles moving through CF sputum; (**D**) Representative images of NP distribution in large airways and lung parenchyma following administration of the respective NPs (red). Image-based quantification of (**E**) coverage of NPs in large airways and (**F**) distribution of NPs in the lung parenchyma. Error bars indicate SD, n = 3 mice/group (>30 sections were analysed per mouse). Two-tailed Student’s *t*-test, ** *p* < 0.01. Reproduced with permission [1]. Copyright 2018, Elsevier.

**Figure 3 pharmaceutics-15-02457-f003:**
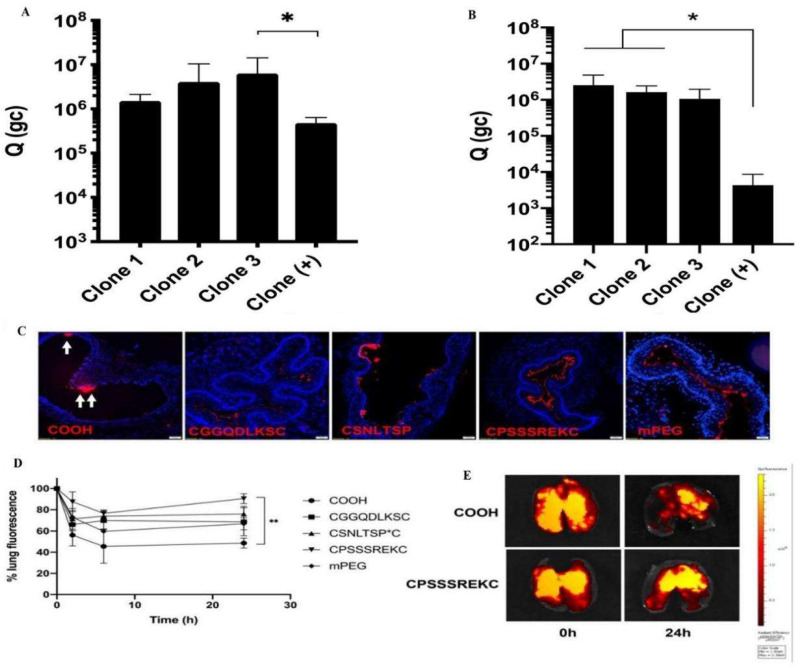
Peptides as nanoparticle surface coatings for the treatment of cystic fibrosis. Amount of phage clones that permeate across (**A**) CF mucus model, (**B**) CF sputum from patients. (**C**) Tracheal distribution at 30 min after administration (*n* = 3). (**D**) Retention in the entire lung over time. The retention percentage is reported as the percentage of the initial fluorescence for each nanoparticle after administration. (**E**) Representative images of the mouse lungs harvested at varying times after intratracheal administration of uncoated and mucus-penetrating CPSSSREKC-conjugated PS nanoparticles. Data represents mean ± SD (n = 3). Kruskal-Wallis test with post hoc Dunn’s multiple comparisons, * *p* < 0.05, ** *p* < 0.01; (**B**) *p* = 0.067 Clone 3 vs. Clone (+). Reproduced with permission [72]. Copyright 2020, Elsevier.

**Figure 4 pharmaceutics-15-02457-f004:**
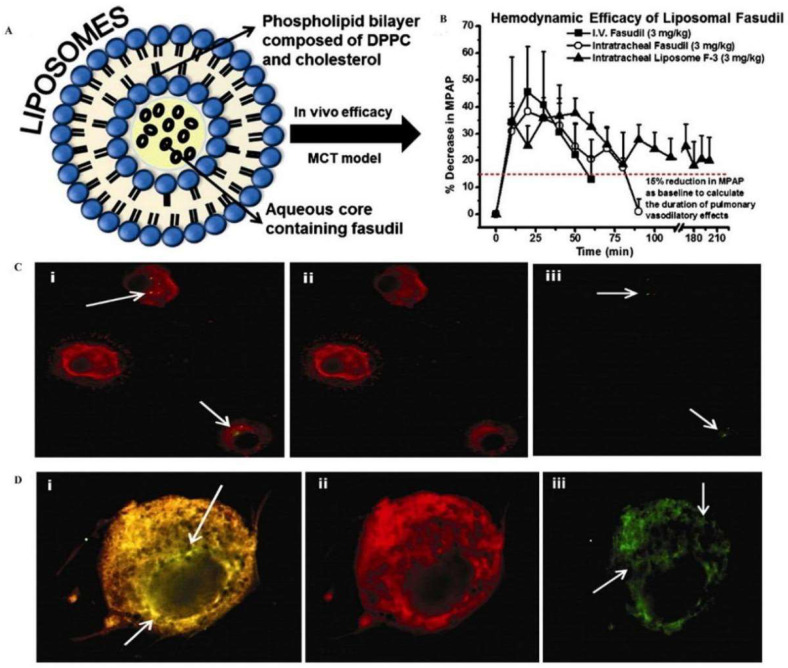
Liposomal fasudil is used in the treatment of pulmonary hypertension. (**A**) Schematic representation of the liposome preparation. (**B**) mean pulmonary arterial pressure (M.P.A.P.) [horizontal dashed line (----) represents a 15% reduction in M.P.A.P.]. Intracellular uptake of liposomes containing fasudil by (**C**) rat alveolar macrophages and (**D**) rat pulmonary arterial smooth muscle cells. Reproduced with permission [49]. Copyright 2013, Elsevier.

**Figure 5 pharmaceutics-15-02457-f005:**
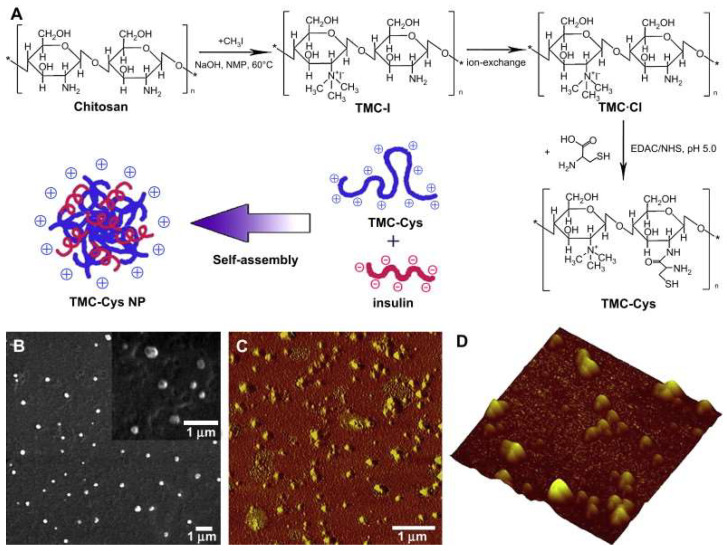
Synthetic route of TMC-Cys and schematic representation of the formation of self-assembled TMC-Cys NP (**A**) and SEM (**B**) as well as AFM (**C**,**D**) images of TMC-Cys(200,30) NP. C corresponded to the phase image, while (**D**) corresponded to the three-dimensional image [91]. Copyright 2009, Elsevier.

**Figure 6 pharmaceutics-15-02457-f006:**
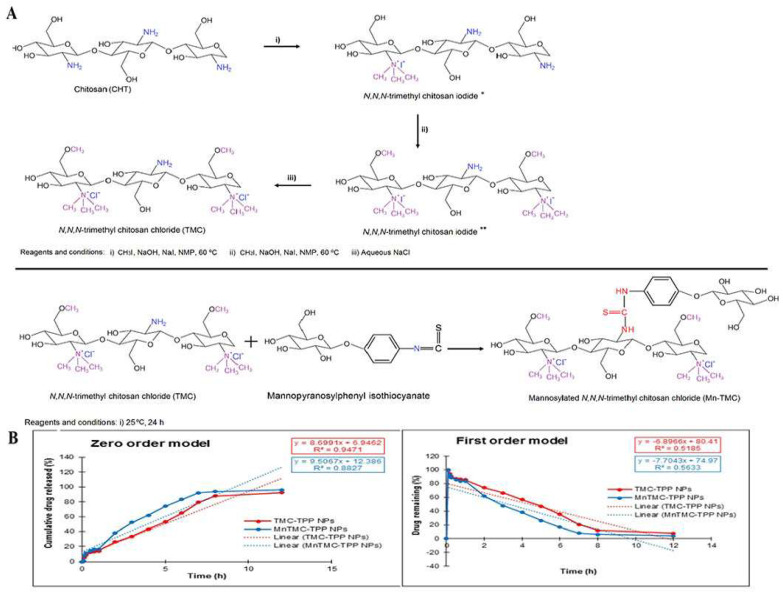
Mannose-anchored N, N, N-trimethyl chitosan nanoparticles for pulmonary administration of etofylline. (**A**) The scheme of synthesis of N, N, N-trimethyl chitosan chloride (TMC) (* product from the first step; ** product from the second step) and mannosylated N, N, N-trimethyl chitosan chloride (MnTMC). (**B**) Mathematical models demonstrating the drug release kinetics from ETO-TMC-TPP and ETO-MnTMC-TPP NPs. Reproduced with permission [55]. Copyright 2020, Elsevier.

**Table 1 pharmaceutics-15-02457-t001:** Summary of nanoparticles for enhanced mucus penetration and retention.

Nanoparticle Type	Formulation Details	Results	Reference
Hydrophilic polymer-modified nanoparticles	Polyethylene glycolized PLGA and polyethyleneimine composite nanoparticles	Effectively prevents nanoparticle uptake by lung macrophages One day after administration, approximately 14% of the polyethylene glycolized nanoparticles were cleared from the lungs, while approximately 37% of the non-polyethylene glycolized particles were eliminated	[45]
	Ph-sensitive methoxy poly(ethylene glycol) (mPEG)-doxorubicin (DOX) coupled nanoparticles	Increased rate and extent of DOX cell internalization DOX released from mPEG–DOX nanoparticles was significantly accelerated in an acidic environment Cytotoxicity: mPEG1K–DOX > free DOX > mPEG2K–DOX ≫ mPEG5K–DOX	[46]
	Poly(acrylic acid) (PAA) and poly(allylamine) (PAM) composite nanoparticles	Obtaining neutral nanoparticles Enhanced mucus penetration	[47]
Cell-penetrating peptide-modified nanoparticles	Phage display libraries as screening tools to identify peptides that promote mucus barrier transport	2.6-fold enhancement in diffusion rate in CF mucus model Reduced affinity for mucin 3-fold increase in cellular uptake of conjugated nanoparticles	[48]
	Direct coupling of peptides to DNA to form nanoparticles	Excellent lung distribution Polyethylene glycolized nanoparticles exhibit excellent colloidal stability 40% PEG-modified nanoparticles exhibit superior transgene expression	[1]
Lipid nanoparticles	Liposomal vesicle encapsulation of fasudil	The embedding efficiency ranged between 68.1 ± 0.8% and 73.6 ± 2.3% 10-fold increase in terminal plasma half-life	[49]
	Insulin/dimyristoylphosphatidylglycerol hydrophobic ion pair, incorporated into a self-nanoemulsifying system	High encapsulation efficiency (70.89%) SNEDDS formulations exhibit increased mucus permeability Protects proteins from enzymatic degradation	[50]
	Lipid polymer hybrid nanoparticles composed of poly(lactic acid-glycolic acid) (PLGA) core and dipalmitoylphosphatidylcholine (DPPC) lipid shell Polyethylene glycol surface coating	PEGylation does not make a difference when mucus barrier properties are dominated by pathology-associated proteins	[51]
Metal nanoparticle	Silver nanoparticles	Levels of hypoxia-inducible factor (HIF)-1α, VEGF, phosphatidylinositol-3 kinase (PI3K) increased Mucous glycoprotein expression (Muc5ac) in lung tissues was substantially decreased	[52]
	Gold nanoparticles	Gold nanoparticles clearly inhibited (70–100%) allergen-induced accumulation of inflammatory cells as well as the production of both pro-inflammatory cytokines and reactive oxygen species Gold nanoparticles prevented airway hyper-reactivity, inflammation and lung remodeling	[53]
Chitosan-modified nanoparticles	Thiol adhesion of anionic polyacrylic acid (PAA) and cationic chitosan (CS) nanoparticles The modification of the polymer was performed by coupling with cysteine (PAA-Cys) and 2-iminothiane (CS-TBA)	Improved mucosal adhesion of thiolated nanoparticles Ranking of particle adhesion ability: CS-TBA > PAA-Cys > CS > PAA CS-TBA has 2-fold higher mucosal adhesion properties than PAA-Cys NP	[54]
	Sodium tripolyphosphate (TPP) as ionic crosslinker Mannose-anchored N,N,N-trimethyl chitosan nanoparticles (TMC) doped with ethoxanthine (ETO) were prepared	The prepared Mn-TMC NPs had a particle size of 223.3 nm, a PDI of 0.490, a ζ potential of −19.1 mV, a drug loading of 76.26 ± 1.2%, and an encapsulation efficiency of 91.75 ± 0.88% The pulmonary bioavailability of TMC-TPP NPs was 4.2 times higher than that of ETO suspension, while the pulmonary bioavailability of MnTMC-TPP NPs was 4.1 times higher than that of ETO suspension	[55]
	N,N,N-trimethyl chitosan nanoparticles (TMC polymers) with different degrees of quaternization were first synthesized	Absorption properties of polymers increase with the degree of quaternization All these polymers led to a mild increase in mucus secretion at pH 4.40. At pH 7.40, only highly quaternized TMC can increase nasal absorption of insulin	[56]
Hyaluronic acid-modified nanoparticles	Encapsulation of salbutamol sulfate (SAS) in dry powder form in inhalable hyaluronic acid (HA) particles	Local albuterol retention in the lungs is more than three times higher than albuterol without HA A reasonable in vitro lung deposition with a fine particle fraction of over 30% Hyaluronic acid allows SAS to have extended release properties and prolong retention time in the lungs The maximum plasma concentration decreased significantly from 2267.7 ng/mL to 566.38 ng/mL	[57]
	Loading of budesonide particles into hyaluronic acid particles using a spray drying process	Longer retention time of HA-modified nanoparticles on the surface of porcine tracheal cannulae Budesonide exhibited significantly prolonged T.max and increased bioavailability in animal models	[58]
Nanoparticles modified with mucolytic agents	Firstly, curcumin-PLGA NP was prepared Multi-drug respirable particles (MP) were prepared with a matrix consisting of N-acetylcysteine (NAC) with antibiotics and the nanoparticles prepared above	The composite nanoparticles embedded with NAC showed strong mucus permeability after 15 min NAC-loaded composite nanoparticles result in reduced TNF-α release 10 mM NAC reduced TNF-α and IL-8 by 45% and 58%, respectively	[59]

**Table 2 pharmaceutics-15-02457-t002:** Summary of patents related to nanoparticles for enhanced mucus permeation and retention.

Patent Name	Inventors	Public Information	Priority Date	Reference
Chitosan-N-arginine nanoparticles, and preparation method and application thereof	HE CHAOLIANG ZHAO DAN	CN115252640 (A) 2022–11–01 CN115252640 (B) 2023–08–29	2022–06–23	[110]
MUCUS-PENETRATING BUDESONIDE NANOSUSPENSION ENEMA FOR LOCAL TREATMENT OF INFLAMMATORY BOWEL DISEASE	HANES JUSTIN [US] ENSIGN LAURA M [US]	US2020306202 (A1) 2020–10–01	2017–07–14	[111]
MUCUS PENETRATING GENE CARRIERS	SUK JUNG SOO [US] HANES JUSTIN SCOT [US]	US2017246320 (A1) 2017–08–31 US10729786 (B2) 2020–08–04	2011–02–08	[112]
NANOPARTICLE FORMULATIONS WITH ENHANCED MUCOSAL PENETRATION	ENSIGN LAURA [US] CONE RICHARD [US]	US2016317459 (A1) 2016–11–03 US9629813 (B2) 2017–04–25	2012–01–19	[113]
Preparation method of powder inhalation for slow-release delivery of COPD (chronic obstructive pulmonary disease) treatment medicine in targeted small airway	LI NAN LI XU	CN115227682 (A) 2022–10–25	2022–07–25	[114]
Acid-sensitive group modified viscous inert nanoparticles as well as preparation method and application of acid-sensitive group modified viscous inert nanoparticles	YU LINLING LIU LING	CN115192734 (A) 2022–10–18	2022–02–22	[115]
Use of nanoparticles for treating respiratory infections associated with cystic fibrosis	NIEDERMEYER WILLIAM	AU2020315585 (A1) 2022–03–03	2019–07–12	[116]
USE OF ALGINATE OLIGOMERS TO ENHANCE THE TRANSLOCATION OF MICRO/NANOPARTICLES ACROSS MUCUS LAYERS	TAGALAKIS ARISTIDES [GB] HART STEPHEN [GB]	US2021030891 (A1) 2021–02–04	2018–03–19	[117]
Polyphosphoester polymer, preparation method thereof, modified porous silicon nanoparticles and preparation method and application of modified porous silicon nanoparticles	LIU WEI RAO RONG	CN109762170 (A) 2019–05–17 CN109762170 (B) 2020–03–31	2019–01–23	[118]

## Data Availability

The study did not report any data.

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
