# Peer review of "Nanoparticle-Mediated Strategies for Enhanced Drug Penetration and Retention in the Airway Mucosa"

_pharmaceutics, 2023, doi:10.3390/pharmaceutics15102457_

Round 1

Reviewer 1 Report

The concept behind the preparation of the present review manuscript titled “Nanoparticle-mediated strategies for enhanced drug penetration and retention in the airway mucosa” by Yan and Sha is very good and well-written. However, the authors need to revise the manuscript very critically. My comments are as follows.

Comment 1. What are the challenges associated with conventional drug delivery systems and how does the nanoparticles-based system enhance drug penetration and retention in the airway mucosa? Kindly discuss this in a separate section.   

Comment 2. Kindly discuss some clinical studies based on nanoparticle-based systems. Further, patents should also be included in the manuscript.

Comment 3. What are the challenges in the clinical translation of nanoparticle-based systems to improve drug penetration and retention in the airway mucosa?

Comment 4. One of the major concerns with nanoparticle-based systems is toxicity after inhalation. Kindly discuss this in detail. 

Reviewer 2 Report

A Review on “Nanoparticle-mediated strategies for enhanced drug penetration and retention in the airway mucosa” is in the scope and well compiled. The author claims to summarise strategies to enhance drug penetration and retention in the airway mucosa mediated by nano-delivery systems, including mucosal permeation systems, mucosal adhesion systems, and enzyme-modified delivery systems

The comments are as follows:

1.       Line 12: Mucus is a complex hydrogel composed… Hydrogel?

2.       Table 1. Summary of nanoparticles for enhanced mucus penetration and retention. In this table, lipid-based nanoparticles, metallic nanoparticles etc. should be incorporated.

3.        As this paper is about retention, details of mucoadhesive polymer for nanoparticles and mechanism with factors should be covered

4.       Clinical study data should be included

5.       Listed of patents on this topic as tabular.

Reviewer 3 Report

The article describes nanoparticle-based drug delivery systems for pulmonary application. Various polymers are discussed, such as mucoadhesieve hialuronic acid and modified chitosans, as well as novel complex drug delivery systems such as SMEDDS, and lipid nano carriers. 

Generally, the review article gives a good oversight about various possibilities of drug delivery targeting the lungs. There are some drawbacks which need to be addressed somehow. 

Firstly, the references could be updated with some more recent ones. There are a number of references that are 10 and more years old.  

Secondly, the references are not only original research articles, but secondary sources, which is not optimal. 

Thirdly, the delivery of nucleic acids into the lungs could be described in more detail. 

Lastly, since there are plenty of similar review articles already published regarding the nanoparticles for pulmonary drug delivery, it is essential to emphasize in abstract in which specific way this particular manuscript brings any novelty compared to the already published ones. 

Specific comments 

Abstract, line 14. Mucus cannot be considered to prevent the absorption of drugs in the lungs. It just represents a barrier for absorption, especially for more lipophilic drugs. 

Line 34. It is better to say that the surface cells beat and relax. 

Line 52. Reformulate “After inhaling drug delivery” 

Lines 77-95. There is no mentioning about the surfactants. 

Line 97. replace um with µm 

Lines 137-140. Please better separate the targeting of mucus and cell penetration as the means to improve absorption. 

Line 175 among other. Explain all abbreviations at the beginning of the article. DPI, FPF, PNA,  

Line 181. According to other sources, lung cancer is not the most common cancer worldwide (https://doi.org/10.2991/jegh.k.191008.001) 

Line 273 – please write hours with word “hour” and not “h” 

Line 337. Reformulate “non-toxic safety” 

Line 360. What kind of mass spectrometer is TA-XTplus? This reviewer has been using various mass spectrometers in the past 23 years and has never heard of such an MS. 

Line 367. Consider replacing the word “fundamental” particles by a more neutral description, like “original”, “control” particles. 

Lines 364-378. Be more specific regarding the targeted mucosa – concentrate either on gastrointestinal tract, or even better, respiratory tract (pulmonary). 

Lines 380-383. Corresponded? 

Line 388. Reformulate or correct the grammar in sentence regarding the weaker hydrogen bonds in chitosan due to quaternized groups. 

Line 415. A threshold or a maximum, plateau? 

Line 436 – the retention time or the residence time? 

Line 443 – Explain the term “phadrug”? 

Line 498-499. Explain why the intestinal pH range is unsuitable for the trypsin action, since this enzyme is supposed to be active in that same intestinal region in vivo. Furthermore, the reference to which the authors are reffering to, is not an original research article, it is a review article.  

Lines 513-514. Please reformulate the sentence since in its present form it doesn’t make any sense. 

The English is generally easy to understand.

Reviewer 4 Report

The review summarizes the strategies for improvement of the drug penetration and retention in the airways mucosa using nanoformulations.

The topic is interesting and extremely important, since the local treatment of the lungs and the administration of active substances through the lungs still present many challenges.

The work summarizes the characteristics and pathophysiology of the airway mucus membrane, as well as their changes in certain disease. Afterwards, the different nanosystems are evaluated.

My  comments would be the following:

1.       Table 1 is confused, the type of nanosystem and the method of production should be separated, the goal and the result are often confused, some parts are incomprehensible without reading the original article, and there are completely meaningless part (eg. „blood samples are collected to measure….”). The rows are mixed.

2.       The lipid-base nanoparticle part is not correct, there is no mention of SLN and NLC systems, why are micelles included here?

3.       In several cases, the sentences are incomprehensible or the wrong words are used: e.g. thiose-modified etc.

There are many hard-to-understand sentences and mistakes in the work, some scientific words are not correct.

Round 2

Reviewer 1 Report

The authors revised the manuscript very carefully. I don't have further comments. 

Reviewer 2 Report

I am satisfied with the addition of data, correction and justification given in the revised manuscript. Thank you for your effort. 

Reviewer 4 Report

The authors addressed my concerns.